# Reference Intervals and Percentiles for Hematologic and Serum Biochemical Values in Captive Bred Rhesus (*Macaca mulatta*) and Cynomolgus Macaques (*Macaca fascicularis*)

**DOI:** 10.3390/ani13030445

**Published:** 2023-01-28

**Authors:** Jaco Bakker, Annemiek Maaskant, Merel Wegman, Dian G. M. Zijlmans, Patrice Hage, Jan A. M. Langermans, Edmond J. Remarque

**Affiliations:** 1Animal Science Department, Biomedical Primate Research Centre, Lange Kleiweg 161, 2288 GJ Rijswijk, The Netherlands; 2Department Population Health Sciences, Animals in Science & Society, Faculty of Veterinary Medicine, Utrecht University, 3584 CM Utrecht, The Netherlands; 3Department of Virology, Biomedical Primate Research Centre, Lange Kleiweg 161, 2288 GJ Rijswijk, The Netherlands

**Keywords:** biochemistry, blood, cynomolgus macaque, hematology, ketamine, medetomidine, normal values, reference intervals, rhesus macaque

## Abstract

**Simple Summary:**

We explored the effect of age, gender, weight-for-height indices, sedation protocol, and housing conditions on the hematologic and serum biochemical values of captive rhesus and cynomolgus macaques. Several blood parameters demonstrated significant and clinically relevant changes in relation to the investigated variables. The results will provide veterinarians and researchers with important reference intervals for evaluating experimental results and health control from rhesus and cynomolgus macaques.

**Abstract:**

Several physiological characteristics and housing conditions are known to affect hematologic and serum biochemical values in macaques. However, the studies that have been conducted either report values calculated based on a small number of animals, were designed specifically to document the effect of a particular condition on the normal range of hematologic and serum biochemical values, or used parametric assumptions to calculate hematologic and serum biochemical reference intervals. We conducted a retrospective longitudinal cohort study to estimate reference intervals for hematologic and serum biochemical values in clinically healthy macaques based on observed percentiles without parametric assumptions. Data were obtained as part of the Biomedical Primate Research Centre (Rijswijk, The Netherlands) health monitoring program between 2018 and 2021. In total, 4009 blood samples from 1475 macaques were analyzed with a maximum of one repeat per year per animal. Data were established by species, gender, age, weight-for-height indices, pregnancy, sedation protocol, and housing conditions. Most of the parameters profoundly affected just some hematologic and serum biochemical values. A significant glucose difference was observed between the ketamine and ketamine-medetomidine sedation protocols. The results emphasize the importance of establishing uniform experimental groups with validated animal husbandry and housing conditions to improve the reproducibility of the experiments.

## 1. Introduction

Rhesus macaques (*Macaca mulatta*) and cynomolgus macaques (*Macaca fascicularis*) are frequently used as biomedical research models [1,2]. Monitoring their wellbeing and physical health is mandatory. Hematologic and serum biochemical values are of great importance in evaluating the individual health status of these macaques. However, many factors such as normal growth and maturation, husbandry, gender, relocation, fasting, and anesthesia can affect hematologic and serum biochemical values in macaques [3,4,5,6,7,8,9,10,11,12,13,14,15,16,17,18,19,20,21,22,23,24,25,26,27,28,29,30,31,32,33,34,35,36,37]. To provide optimal healthcare and to advance our understanding of macaque models of human disease, it is essential to determine the correct reference intervals to avoid inaccurate conclusions from studies on macaques. In addition, macaques are a prey species and therefore tend to mask disease until it is very severe. This can complicate attempts to evaluate the health status or effect of a treatment, vaccination, or therapy on macaques during experimental research. The early detection of underlying disease is especially important when using animal models for human disease to avoid confounding.

Currently, limited variables are investigated for assessing the reference intervals for macaques (mostly age and gender); reference intervals are calculated from relatively small numbers, and parametric assumptions are used [3,4,5,6,7,8,9,10,11,12,13,14,15,16,17,18,19,20,21,22,23,24,25,26,27,28,29,30,31,32,33,34,35,36,37]. Therefore, hematologic (HER) and serum biochemical (CER) values from a large cohort of domestically bred rhesus and cynomolgus macaques without overt clinical signs of disease were analyzed in this study. As many hematologic and serum biochemical values are not normally distributed, data are presented as observed percentiles without parametric assumptions. We hypothesize that significant differences in HER and CER values in relation to age, gender, weight-for-height indices, used sedation protocol, pregnancy, and housing conditions exist. Our study provides veterinarians and researchers with a complete set of reference intervals for domestically reared macaques.

## 2. Materials and Methods

All data were obtained retrospectively from the electronic health and medical records of animals that were housed at the Biomedical Primate Research Centre (BPRC, Rijswijk, Netherlands) between 2018 and 2021. All animals were housed in accordance with Dutch law and international ethical and scientific standards and guidelines (EU Directive 63/2010). All procedures and husbandry were compliant with the above standards and legislations. The BPRC is accredited by the Association for Assessment and Accreditation of Laboratory Animal Care (AAALAC).

The BPRC houses an outbred breeding colony of rhesus macaques (*Macaca mulatta*) and cynomolgus macaques (*Macaca fascicularis*), consisting of approximately 1000 rhesus and 150 cynomolgus macaques. The colony was formed around 1975 and consisted initially of captive-bred macaques obtained from various accredited suppliers. Later, new breeding lines were introduced on several occasions to maintain the outbred character of the colony.

All animals underwent annual health evaluations that included weight-for-height indices assessment [38] and blood sampling for HER and CER evaluation. In addition, a thorough physical examination including a pregnancy test was performed by a veterinarian.

The animals were not considered to be specific pathogen-free as they were potentially infected with other common subclinical viral pathogens, including simian foamy virus. The colony was negative for *Mycobacterium tuberculosis*, *Salmonella enterica Typhi*, and *Shigella dysenteriae* at the time of sampling.

Qualified animal caretakers observed all animals for injuries, illness, and fecal consistency at least twice daily. Abnormalities were reported to a veterinarian. Parturitions and stillbirths were recorded, and ultrasonography was used to assist in the judgment of the animal being pregnant at the time of blood sampling.

### 2.1. Housing Condition

The majority of the colony was born and bred in our breeding groups with outdoor access. To perform animal experiments, the macaques were relocated to permanent indoor housing. Therefore, all animals that resided in indoor housing all had a history of outdoor access. The specifics of the housing conditions are detailed below.

#### 2.1.1. Outdoor Access

The animals were housed socially in open enclosures holding naturalistic family groups of between 20 and 30 macaques. The enclosures consisted of freely accessible indoor and outdoor enclosures [39]. Both indoor (75 m^2^, 2.85 m high) and outdoor enclosures (208 m^2^, 3.1 m high) consisted of several compartments and visual barriers. Outdoor enclosures were covered by a galvanized wire mesh and had roofed areas, windbreaks, and shaded areas. The floors in the indoor enclosures were provided with wood fiber bedding (Lignocel ¾ Grade x, J. Rettenmaier & Sohne GmbH + CO, Rosenberg, Germany), whereas outdoor enclosures had sand bedding. The cleaning procedures of the outdoor enclosures consisted of removing feces on a regular basis. As for the indoor enclosures, high-pressure water cleaning, including disinfection (Anistel Surface disinfectant, Tristel Solutions Limited, Cambridgeshire, UK), was performed monthly. Following disinfection, the enclosures were rinsed with clean water, and the floor was wiped dry. After allowing for a 30–40 min air drying period, new bedding was provided. Environmental enrichment consisted of several climbing structures, beams, fire hoses, and sitting platforms. The indoor enclosure’s temperature was at minimum 18 °C for rhesus and 21 °C for cynomolgus macaques, with a 12 h light–dark cycle, and with six air changes per hour. Concerning outdoor enclosures, the atmospheric temperature ranged from approximately −5 to 37 °C throughout the year with a maritime climate. The animals were fed commercial monkey pellets (Ssniff, Soest, Germany) supplemented with limited amounts of fruit, vegetables, or grain mixtures in the afternoon. Enrichment-containing food was provided regularly. Municipal water was available ad libitum, provided by automatic water dispensers.

#### 2.1.2. Permanent Indoor Housing but with a History of Outdoor Access (see Section 2.1.1)

Pair-housed animals used in various research programs were housed in indoor cages measuring D 2 × W 2 × H 3 m. The room temperature was 20 ± 2 °C, with six air changes per hour, with a 12 h light–dark cycle and relative humidity of 50 ± 10%. The wood fiber bedding of the enclosures was replaced weekly; one week without additional cleaning procedures, one week after rinsing with tap water, and one week after high-pressure water cleaning, including disinfection (Anistel Surface disinfectant). Enrichment such as mirrors and toys were provided weekly. The animals were fed commercial monkey pellets (Ssniff, Soest, Germany), and limited amounts of fruits or vegetables were provided twice daily. All animals also received additional food enrichment items daily. Municipal water was available ad libitum, provided by automatic water dispensers.

### 2.2. Data Collection and Analysis

Blood samples were obtained during annual veterinary examinations between 2018–2021. These annual exams were performed distributed over the year, with an aimed interval of approximately 12 months for each animal. The animals that were included in the analysis did not present clinical signs of diseases based on daily care and observations. The standard health program did not include preventative anti-parasitic treatment. All study animal pedigrees and birth dates were known. Animals were included in the annual veterinary examinations when they were aged 6 and 9 months and over, for rhesus and cynomolgus macaques, respectively, resulting in an age range between 0.5–27.7 and 0.6–26.7 years for rhesus and cynomolgus macaques, respectively. Table 1 and Table 2 shows the animal details by the number of repeats by gender and species.

The macaques were fasted overnight (16 h) prior to sample collection while water was maintained available throughout. All blood samples were collected in the morning.

An effort was made to minimize stress during the capture procedure before sedation. As our institute is well-experienced and progressive in refinement, a continuous training program was developed to establish cooperation during the capture procedure in the group-housed macaques. During this training program, the animals were trained to voluntarily enter an individual squeeze cage. After an intramuscular injection of the sedative, the body weight was recorded, and blood samples were obtained by qualified caretakers approximately within 20 min. The collection site was sterilized with alcohol 70% and 1 mL EDTA, and 2.5 mL clothed blood samples were collected from a femoral vein by using a 20-gauge Vacuette needle and hub. The samples for HER were collected in EDTA tubes (Greiner Bio-One GmbH, Kremsmünster, Austria) and mixed by inversion, whereas the samples for CER were collected in tubes without anticoagulants (Greiner Bio-One GmbH, Kremsmünster, Austria) and allowed to clot at room temperature for one hour. HER samples with clots or insufficient blood for the anticoagulant were discarded, and hemolytic serum samples were excluded from the CER analysis. Prior to processing and analysis HER samples were once again mixed by inversion and the CER samples were centrifuged at 3000 r.p.m. for 10 minutes. All values were determined on-site at the BPRC within 24 h after sampling.

HER was determined by using a Sysmex XT2000iV (Sysmex BV, Etten-Leur, the Netherlands). The hematologic parameters measured included: red blood cell count (RBC, 10^12^/L), hemoglobin (HGB, mmol/L), hematocrit (HCT, L/L), mean corpuscular volume (MCV, fL), mean corpuscular hemoglobin, (MCH, amol), mean corpuscular hemoglobin concentration (MCHC, mmol/L), platelet count (PLT, 10^9^/L), platelet distribution width (PDW, fL), mean platelet volume (MPV, fL), platelet larger cell ratio (P-LCR, %), plateletcrit (PCT, %), white blood cell count (WBC, 10^9^/L), neutrophil count (Neut, 10^9^/L), lymphocyte count (Lymph, 10^9^/L), monocyte count (Mono, 10^9^/L), eosinophil count (Eo, 10^9^/L), and basophil count (Baso, 10^9^/L).

CER was determined by using a Cobas Integra 400 plus analyzer (F. Hoffmann-La Roche Ltd., Switzerland): albumin (ALB, g/L), total protein (TP, g/L), alkaline phosphatase (ALP, U/L), alanine aminotransferase (ALT, U/L), aspartate aminotransferase (AST, U/L), lactate dehydrogenase (LDH, U/L), gamma-glutamyltransferase (GGT, U/L), total bilirubin (TBIL, umol/L), cholesterol (Chol, mmol/L), chloride (Cl, mmol/L), bicarbonate (BIC, mmol/L), iron (Fe, umol/L), potassium (K, mmol/L), sodium (Na, mmol/L), phosphate (Phos, mmol/L), calcium (Ca, mmol/L), urea (URE, mmol/L), and creatinine (Cre, umol/L).

Both machines were calibrated routinely every 6 months by a service professional.

### 2.3. Investigated Variables

The investigated variables were:Age: macaques < 4 and ≥4 years of age were compared. The age of 4 years was selected as macaques are considered to reach sexual maturity at the age of 4 years;Gender: males versus females;Weight-to-height ratio assessed by weight-for-height indices (WHI). The best measure of relative adiposity and explored the boundaries of overweight and underweight in captive group-housed rhesus and cynomolgus macaques is a species-specific WHI with height to the power of 3.0 (rhesus macaques) and 2.7 (cynomolgus macaques) as it depended least on height and showed high correlation with other relative adiposity measures [38]. For rhesus macaques, a WHI > 67, and for cynomolgus macaques, a WHI > 62 were set as overweight. Only adult animals were included in this comparison;Sedation protocol: ketamine (years 2018–2019) and a combination of ketamine with medetomidine (years 2020–2021) were compared, both administered intramuscularly. The animals were sedated with a predetermined dose, based on their last noted weight, resulting in a dose of approximately: (A) 10 mg/kg ketamine (ketamine 10%; Alfasan Nederland BV, Woerden, Netherlands); (B) ketamine (10 mg/kg, ketamine 10%; Alfasan Nederland BV, Woerden, Netherlands) combined with medetomidine (0.05 mg/kg, Sedastart; AST Farma BV, Oudewater, Netherlands). After finishing the procedures, the animals were allowed to return to their enclosure, and atipamezole (0.25 mg/kg, Sedastop; AST Farma BV, Oudewater, Netherlands) was administered intramuscularly;Outdoor access versus permanent indoor housing with a history of outdoor access (Section 2.1.1 versus Section 2.1.2). Only rhesus macaques ≥ 4 years of age were included. The cynomolgus macaques were not included because of the small sample size;Pregnancy: 57 rhesus and 40 cynomolgus macaques were determined to be pregnant by a veterinarian at the moment of blood sampling. The values were compared with nonpregnant females (1407 rhesus and 353 cynomolgus macaques).

### 2.4. Statistics

The reference intervals for a laboratory parameter are defined by the boundaries of the parameter that include 95% of all observations in a population. It is assumed that 95% of a population is normal. The 5% considered not normal are the 2.5% smaller than the lower reference value and the 2.5% greater than the upper reference value. Because most laboratory parameters are not normally distributed, the estimation of the 2.5 and 97.5% percentiles should not be performed when using parametric methods (i.e., Arithmetic Mean ± 1.96 ∗ Standard Deviations). Therefore, the 2.5 and 97.5% percentiles are calculated using nonparametric methods. We used the criteria outlined by Lahti et al. (2002) to decide whether to establish different reference intervals for age and gender groups; the criteria for not partitioning are that >0.9% and <4.1% of the group distributions should be outside the 2.5 and 97.5 percentiles of the common distribution [40]. Nonparametrically estimated reference intervals should be based on group sizes greater than 120; therefore, partitioning resulting in groups with less than 120 observations was not used to define reference intervals [41].

To be able to compare the magnitude of the differences between the groups, between-group differences are expressed as the percentual difference of the medians (hereafter referred to as Delta), which is calculated as 100 ∗ (Median Group 1 − Median Group 2)/Mean (Median Group 1 and Median Group 2). Absolute delta values greater than 5% were considered clinically relevant. The statistical significance of between-group differences was evaluated using nonparametric tests (Mann–Whitney for independent and Wilcoxon’s signed rank test for paired observations), and *p* values < 0.001 were considered statistically significant.

## 3. Results

Only absolute delta values greater than 5% that were also significant (*p* < 0.001) were considered clinically relevant and will be discussed below. Table 3, Table 4, Table 5 and Table 6 shows the HER and CER intervals and percentiles, excluding glucose, for rhesus and cynomolgus macaques.

### 3.1. Temporal Trends and Sedation Protocol

The most prominent finding was a significant increase in glucose levels between 2018–2019 and 2020–2021 in all ages, genders, and both species (12.2% young females; 13.6% adult females; 10.2% young males; 10.2% adult males in rhesus macaques and 14.9% young females; 15.8% adult females; 15.8% young males; 12.1% adult males in cynomolgus macaques; *p* < 0.000001) (Appendix A). This is a clear indication that the sedation protocol, which changed in 2020 from ketamine to ketamine-medetomidine, influences glucose levels. Therefore, in Table 3, Table 4, Table 5 and Table 6, the reference intervals are presented without glucose. The glucose reference intervals are shown in separate tables (Table 7 and Table 8). Moreover, higher ALP levels in male rhesus macaques in 2019 compared to 2018 (10.8%; *p* < 0.000001) and 2020 compared to 2019 (10.5% *p* < 0.000001) were observed.

We used the criteria outlined by Lahti et al. (2002) to decide whether to establish different reference intervals for age and gender groups; the criteria for not partitioning were that >0.9% and <4.1% of the group distributions should be outside the 2.5 and 97.5 percentiles of the common distribution. Our results showed that reference values for K, Na, and MCHC values can be combined for all age groups and both genders in rhesus macaques (Table 9). Eo can be combined in cynomolgus macaques in all age groups and both genders (Table 9). TBIL can be combined in cynomolgus macaques for both ages but not per gender. In rhesus macaques, Cl, K, PLT, P-LCR, Neut, and Mono can be combined for both ages but not per gender (Table 10). ALT, AST, LDH, Chol, BIC, MCV, MCH, MCHC, PDW, and WBC can be combined for rhesus macaques for gender but not age groups (Table 11).

Concerning glucose, for rhesus macaques, reference intervals can be combined for both genders and both ages, using the reference intervals for sedation protocols A and B (Table 12).

**Table 3 animals-13-00445-t003:** Serum biochemical reference intervals and percentiles in the rhesus macaques excluding glucose.

Labtest (Unit)	Gender	Age	Count	Mean	SD	Median	P25	P75	RefLow	RefUpp	Min	Max
ALB (g/L)	F	<4	571	41.89	3.38	42.01	40.09	43.87	34.42	48.24	20.63	49.81
	F	≥4	1463	40.21	3.78	40.58	38.66	42.58	30.27	46.23	20.06	51.60
	M	<4	496	42.25	2.94	42.32	40.72	44.21	36.66	47.27	24.31	50.89
	M	≥4	486	42.91	3.16	43.09	41.22	44.78	35.77	48.55	22.95	51.54
TP (g/L)	F	<4	570	61.64	3.94	61.20	59.10	63.80	54.63	69.79	43.70	77.10
	F	≥4	1464	63.57	4.64	63.60	60.90	66.20	54.16	73.20	38.90	81.20
	M	<4	496	61.37	3.53	61.50	59.20	63.53	55.00	68.52	41.90	73.70
	M	≥4	486	63.78	3.71	63.65	61.30	66.20	57.04	71.00	49.00	79.90
ALP (U/L)	F	<4	571	559.2	224.6	524.2	392.2	687.7	236.2	1105.8	138.4	1390.4
	F	≥4	1466	172.4	77.8	152.6	121.5	202.4	79.2	374.5	48.9	787.7
	M	<4	496	668.7	199.8	643.6	531.2	780.6	361.2	1173.1	122.6	1514.3
	M	≥4	485	355.2	213.7	337.5	162.0	495.2	81.6	858.5	57.5	1056.4
ALT (U/L)	F	<4	573	30.12	10.74	28.70	23.40	35.00	14.35	58.68	9.20	84.50
	F	≥4	1467	35.71	28.66	28.50	21.50	40.10	12.00	103.40	4.80	383.80
	M	<4	496	32.85	14.27	30.40	25.20	37.20	16.19	65.11	6.50	177.90
	M	≥4	486	39.44	28.03	29.85	21.63	46.05	12.26	124.79	7.80	181.60
AST (U/L)	F	<4	573	39.74	16.17	37.00	30.10	46.00	18.97	78.59	12.40	183.60
	F	≥4	1466	31.41	17.36	28.50	23.53	35.30	15.40	61.77	8.60	347.60
	M	<4	496	42.52	18.54	38.60	32.78	47.90	20.14	83.95	14.10	215.20
	M	≥4	486	31.13	11.67	29.35	24.43	35.05	15.52	61.21	11.60	159.60
GGT (U/L)	F	<4	572	74.90	24.01	72.60	56.10	90.80	37.33	125.94	25.00	169.20
	F	≥4	1465	46.13	11.48	45.00	39.10	52.60	26.27	70.37	5.40	125.90
	M	<4	496	90.33	23.17	88.95	75.20	102.10	46.82	141.02	31.50	179.40
	M	≥4	486	70.83	23.01	67.50	53.83	83.98	33.70	124.49	26.60	170.80
LDH (U/L)	F	<4	571	461.1	201.7	414.0	333.0	540.0	215.0	968.6	151.0	1733.0
	F	≥4	1464	425.2	206.7	380.0	296.0	497.3	201.0	844.3	142.0	2807.0
	M	<4	496	477.7	178.7	451.0	346.0	576.3	239.0	889.6	166.0	1395.0
	M	≥4	486	436.7	193.6	394.0	292.3	534.0	189.2	901.7	158.0	1447.0
TBIL (µmol/L)	F	<4	561	1.31	0.82	1.10	0.70	1.80	0.10	3.39	0.00	4.60
	F	≥4	1389	1.12	0.85	0.90	0.50	1.50	0.10	3.13	0.00	11.80
	M	<4	493	1.39	0.82	1.20	0.80	1.80	0.20	3.50	0.00	4.20
	M	≥4	478	1.18	0.68	1.00	0.70	1.50	0.20	2.80	0.00	4.40
Chol (mmol/L)	F	<4	570	3.43	0.73	3.34	2.96	3.80	2.33	5.30	0.86	6.11
	F	≥4	1465	3.43	0.63	3.43	3.02	3.81	2.12	4.78	0.99	6.20
	M	<4	496	3.52	0.69	3.43	3.03	3.95	2.43	5.12	2.23	6.35
	M	≥4	485	3.21	0.58	3.17	2.81	3.57	2.18	4.50	1.90	5.08
Cl (mmol/L)	F	<4	573	107.4	2.8	107.3	105.7	109.2	102.2	112.6	93.2	114.6
	F	≥4	1463	107.5	2.8	107.5	105.6	109.4	101.8	112.9	96.8	116.4
	M	<4	495	106.6	2.9	106.6	104.9	108.4	101.2	112.1	89.5	114.1
	M	≥4	485	105.4	2.5	105.5	103.8	107.0	100.9	110.6	96.7	118.6
BIC (mmol/L)	F	<4	572	22.44	3.69	22.75	20.70	24.80	14.53	28.17	0.80	31.70
	F	≥4	1466	23.45	3.77	23.70	21.40	25.70	15.34	30.30	0.10	36.50
	M	<4	496	22.76	3.72	22.75	20.60	25.20	15.34	30.42	4.10	34.00
	M	≥4	486	24.80	3.79	24.90	22.30	27.10	17.52	31.97	0.70	34.40
Fe (mmol/L)	F	<4	572	17.26	5.86	16.96	13.20	21.05	6.55	28.62	0.79	40.88
	F	≥4	1465	18.83	6.47	18.63	14.40	22.94	6.53	31.26	1.90	62.45
	M	<4	496	18.14	5.83	17.80	14.01	21.71	8.15	31.12	4.08	37.03
	M	≥4	486	23.37	5.75	23.09	19.67	27.18	12.75	34.47	5.43	41.31
K (mmol/L)	F	<4	564	3.49	0.39	3.44	3.26	3.66	2.92	4.33	1.96	6.07
	F	≥4	1437	3.58	0.36	3.57	3.36	3.76	2.94	4.32	1.24	6.38
	M	<4	490	3.54	0.35	3.51	3.33	3.70	2.96	4.42	1.84	5.65
	M	≥4	485	3.70	0.33	3.69	3.50	3.89	3.17	4.38	2.50	5.39
Na (mmol/L)	F	<4	572	146.1	2.7	146.2	144.6	147.8	140.9	151.0	128.0	154.6
	F	≥4	1463	145.6	2.7	145.7	143.9	147.4	139.8	150.6	130.5	155.0
	M	<4	496	146.3	3.0	146.7	144.7	148.0	140.9	151.2	127.2	154.8
	M	≥4	485	146.2	2.5	146.5	144.7	147.8	141.1	150.7	132.6	151.9
Phos (mmol/L)	F	<4	571	1.74	0.38	1.72	1.50	1.97	0.98	2.53	0.69	3.28
	F	≥4	1464	1.12	0.39	1.07	0.86	1.32	0.54	2.00	0.30	4.26
	M	<4	496	1.87	0.32	1.87	1.65	2.06	1.27	2.52	0.89	3.17
	M	≥4	486	1.45	0.36	1.46	1.20	1.70	0.75	2.17	0.30	2.51
Ca (mmol/L)	F	<4	570	2.40	0.14	2.40	2.31	2.50	2.14	2.68	1.34	2.83
	F	≥4	1466	2.32	0.13	2.32	2.22	2.41	2.06	2.59	1.89	3.04
	M	<4	496	2.40	0.13	2.39	2.30	2.48	2.18	2.65	2.09	2.97
	M	≥4	484	2.39	0.11	2.39	2.32	2.46	2.16	2.63	1.96	2.72
URE (mmol/L)	F	<4	571	7.37	1.47	7.24	6.38	8.23	4.69	10.63	3.22	13.34
	F	≥4	1465	6.53	1.41	6.39	5.53	7.34	4.12	9.77	3.32	12.41
	M	<4	496	7.57	1.65	7.34	6.58	8.40	4.87	11.05	3.52	22.50
	M	≥4	486	6.74	1.32	6.59	5.85	7.50	4.43	9.72	2.68	11.53
Cre (mmol/L)	F	<4	571	57.22	13.58	56.30	47.90	64.50	36.10	90.04	23.40	136.60
	F	≥4	1461	77.32	14.49	75.90	67.20	85.20	53.51	109.03	40.00	188.40
	M	<4	495	60.04	14.29	58.60	49.00	69.05	34.88	90.14	29.30	131.20
	M	≥4	486	86.76	16.26	84.60	75.10	95.70	60.28	126.99	48.80	146.20

**Table 4 animals-13-00445-t004:** Hematologic reference intervals and percentiles in the rhesus macaques.

Labtest (Unit)	Gender	Age	Count	Mean	SD	Median	P25	P75	RefLow	RefUpp	Min	Max
RBC (10^12^/L)	F	<4	570	5.51	0.37	5.51	5.28	5.75	4.76	6.24	4.20	6.88
	F	≥4	1467	5.46	0.43	5.47	5.22	5.72	4.52	6.27	2.70	6.87
	M	<4	487	5.62	0.40	5.57	5.37	5.87	4.97	6.44	3.55	7.32
	M	≥4	486	5.71	0.37	5.71	5.47	5.94	5.03	6.46	4.35	6.73
HGB (mmol/L)	F	<4	570	8.01	0.49	8.00	7.70	8.30	7.00	9.00	6.20	9.50
	F	≥4	1467	8.01	0.59	8.10	7.70	8.40	6.60	9.00	4.30	10.10
	M	<4	487	8.13	0.51	8.10	7.80	8.50	7.30	9.08	5.10	9.50
	M	≥4	486	8.49	0.48	8.50	8.20	8.80	7.50	9.30	6.30	10.00
HCT (L/L)	F	<4	570	0.38	0.02	0.38	0.36	0.39	0.33	0.42	0.30	0.45
	F	≥4	1467	0.38	0.03	0.38	0.37	0.40	0.32	0.43	0.23	0.49
	M	<4	487	0.38	0.02	0.38	0.37	0.40	0.34	0.42	0.24	0.46
	M	≥4	486	0.40	0.02	0.40	0.39	0.41	0.36	0.45	0.32	0.46
MCV (fL)	F	<4	570	68.52	2.67	68.60	66.93	70.20	62.73	73.77	58.30	75.50
	F	≥4	1468	69.60	2.83	69.50	67.70	71.50	64.20	74.90	58.70	83.30
	M	<4	488	67.65	2.73	67.60	65.60	69.50	62.65	72.96	56.10	75.30
	M	≥4	486	69.96	2.84	69.90	68.10	71.90	64.60	75.58	61.00	78.90
MCHC (amol)	F	<4	570	1454.9	61.7	1455.0	1419.3	1492.8	1325.6	1568.5	1170.0	1643.0
	F	≥4	1468	1470.8	74.3	1470.0	1431.0	1508.0	1353.0	1587.3	1039.0	2791.0
	M	<4	487	1447.1	60.5	1448.0	1409.5	1490.5	1321.2	1552.4	1166.0	1635.0
	M	≥4	486	1488.6	60.9	1493.0	1454.0	1528.0	1358.6	1592.8	1204.0	1636.0
MCHC (mmol/L)	F	<4	570	21.24	0.53	21.30	20.90	21.60	20.30	22.20	16.40	22.80
	F	≥4	1468	21.14	0.67	21.20	20.80	21.40	20.07	22.00	15.40	33.60
	M	<4	487	21.40	0.57	21.50	21.10	21.70	20.42	22.30	16.30	22.80
	M	≥4	486	21.28	0.55	21.30	21.00	21.60	20.30	22.18	17.20	22.80
PLT (10^9^/L)	F	<4	569	330.9	73.7	335.0	287.0	376.0	162.8	470.3	55.0	589.0
	F	≥4	1467	312.4	74.7	308.0	265.0	358.0	178.0	464.6	66.0	726.0
	M	<4	486	319.2	76.3	323.0	273.0	366.0	151.4	454.7	122.0	628.0
	M	≥4	485	298.5	68.2	302.0	256.0	346.0	150.0	419.9	69.0	511.0
PDW (fL)	F	<4	541	13.47	2.16	13.10	11.90	14.70	10.16	18.45	9.10	21.80
	F	≥4	1450	12.68	1.94	12.50	11.30	13.80	9.50	17.30	8.40	21.00
	M	<4	467	12.87	2.18	12.40	11.30	14.10	9.57	18.63	9.00	21.60
	M	≥4	478	12.35	1.83	12.10	11.10	13.30	9.10	17.00	8.60	19.40
MPV (fL)	F	<4	541	11.46	0.92	11.40	10.80	12.20	9.80	13.15	9.10	14.20
	F	≥4	1450	11.12	0.95	11.10	10.40	11.80	9.20	12.90	8.60	13.80
	M	<4	467	11.12	0.97	11.10	10.40	11.80	9.30	13.00	8.60	14.50
	M	≥4	478	10.97	0.96	10.90	10.30	11.60	9.00	12.80	8.50	13.80
P-LCR (%)	F	<4	541	36.07	6.68	36.20	31.00	41.40	22.70	48.55	17.10	56.60
	F	≥4	1450	33.68	7.30	34.00	28.40	39.00	18.40	47.07	13.50	53.70
	M	<4	467	33.44	7.18	33.30	28.30	38.70	19.20	46.93	15.10	58.10
	M	≥4	478	32.39	7.32	32.30	27.60	37.70	16.99	45.81	12.50	54.30
PCT (%)	F	<4	541	0.383	0.065	0.380	0.340	0.420	0.236	0.510	0.150	0.620
	F	≥4	1449	0.345	0.066	0.340	0.300	0.380	0.220	0.478	0.100	0.660
	M	<4	466	0.358	0.065	0.360	0.320	0.400	0.217	0.473	0.140	0.610
	M	≥4	477	0.327	0.059	0.330	0.290	0.370	0.200	0.430	0.110	0.550
WBC (10^9^/L)	F	<4	570	13.55	4.79	13.03	10.02	16.22	6.13	23.73	4.57	39.88
	F	≥4	1468	12.11	5.33	11.23	8.00	15.47	4.47	24.21	1.16	35.79
	M	<4	488	12.20	4.33	11.61	9.03	14.89	5.42	22.54	3.25	26.59
	M	≥4	486	9.46	4.03	8.43	6.77	11.23	4.43	20.39	2.65	27.32
Neut (10^9^/L)	F	<4	568	9.25	4.80	8.53	5.69	11.90	2.37	19.03	1.36	31.63
	F	≥4	1458	9.32	5.22	8.31	5.08	12.73	2.20	21.08	0.89	32.62
	M	<4	481	7.97	4.30	7.36	4.46	10.57	1.93	18.18	0.93	23.75
	M	≥4	478	6.15	3.86	5.11	3.51	7.63	1.78	17.44	1.21	25.63
Mono (10^9^/L)	F	<4	570	0.660	0.313	0.610	0.443	0.810	0.220	1.420	0.070	2.500
	F	≥4	1467	0.636	0.313	0.580	0.410	0.780	0.230	1.393	0.090	3.010
	M	<4	488	0.578	0.257	0.540	0.390	0.720	0.210	1.226	0.100	1.620
	M	≥4	485	0.559	0.245	0.510	0.380	0.690	0.220	1.119	0.040	1.680
Lymph (10^9^/L)	F	<4	570	3.456	1.620	3.060	2.333	4.260	1.360	7.668	0.870	11.770
	F	≥4	1467	2.067	0.968	1.870	1.390	2.535	0.790	4.519	0.140	8.460
	M	<4	488	3.523	1.541	3.270	2.370	4.265	1.351	7.598	0.940	9.440
	M	≥4	485	2.644	1.152	2.450	1.820	3.250	0.873	5.648	0.470	7.250
Eo (10^9^/L)	F	<4	554	0.148	0.188	0.070	0.010	0.220	0.000	0.691	0.000	1.090
	F	≥4	1415	0.092	0.128	0.040	0.000	0.130	0.000	0.460	0.000	0.830
	M	<4	470	0.138	0.169	0.070	0.010	0.210	0.000	0.650	0.000	0.900
	M	≥4	462	0.114	0.150	0.055	0.000	0.168	0.000	0.580	0.000	0.910
Baso (10^9^/L)	F	<4	569	0.019	0.014	0.020	0.010	0.020	0.000	0.050	0.000	0.180
	F	≥4	1454	0.011	0.008	0.010	0.010	0.010	0.000	0.030	0.000	0.080
	M	<4	488	0.016	0.011	0.010	0.010	0.020	0.000	0.040	0.000	0.090
	M	≥4	472	0.010	0.008	0.010	0.010	0.010	0.000	0.030	0.000	0.060

**Table 5 animals-13-00445-t005:** Serum biochemical reference intervals and percentiles in the cynomolgus macaques excluding glucose.

Labtest (Unit)	Gender	Age	Count	Mean	SD	Median	P25	P75	RefLow	RefUpp	Min	Max
ALB (g/L)	F	<4	204	41.22	4.10	41.67	39.27	44.05	29.06	46.97	24.15	47.47
	F	≥4	392	38.00	4.16	38.31	36.01	40.65	28.39	45.58	23.45	48.72
	M	<4	207	41.96	2.27	41.91	40.42	43.60	37.74	46.23	35.59	48.96
	M	≥4	128	40.50	3.67	40.85	38.14	43.27	33.02	47.03	28.33	48.03
TP (g/L)	F	<4	203	62.58	4.24	62.30	59.75	65.40	54.74	71.36	47.30	73.90
	F	≥4	391	67.63	5.46	67.70	64.05	71.00	57.60	79.12	50.30	83.50
	M	<4	207	62.63	3.61	62.50	59.90	64.90	55.94	69.92	53.00	73.10
	M	≥4	128	65.99	4.04	66.45	63.38	68.65	57.94	73.38	54.80	75.60
ALP (U/L)	F	<4	205	645.6	286.4	631.5	445.9	790.0	185.3	1326.7	91.1	1617.8
	F	≥4	392	187.4	93.1	170.3	129.9	220.3	78.1	411.9	56.7	899.7
	M	<4	207	757.2	243.3	705.7	581.1	902.8	385.9	1378.4	284.6	1544.2
	M	≥4	128	296.1	263.2	225.9	103.5	402.9	45.0	1085.2	42.6	1551.1
ALT (U/L)	F	<4	205	36.99	15.01	35.50	28.30	44.60	11.80	77.76	5.90	104.20
	F	≥4	392	42.03	32.74	34.35	24.95	49.23	13.60	112.64	5.30	377.00
	M	<4	207	40.86	17.81	38.30	30.05	47.85	18.04	81.04	12.10	145.40
	M	≥4	128	33.37	17.78	28.60	22.25	39.15	12.35	91.54	8.60	101.20
AST (U/L)	F	<4	205	45.90	16.48	44.10	34.10	52.00	23.92	91.39	13.10	135.50
	F	≥4	392	40.59	15.09	37.80	30.98	48.43	20.23	81.75	15.20	123.80
	M	<4	207	48.33	30.16	45.10	35.35	52.95	23.84	91.12	19.10	420.20
	M	≥4	127	35.53	13.19	32.60	28.25	40.30	18.34	71.34	14.70	119.80
GGT (U/L)	F	<4	204	86.87	39.04	81.70	61.98	102.35	38.44	156.75	24.80	401.20
	F	≥4	392	57.78	31.97	52.35	43.28	60.98	25.96	146.09	14.40	338.30
	M	<4	207	101.61	27.56	101.50	81.90	119.55	55.98	159.98	37.00	182.60
	M	≥4	128	81.48	38.84	74.25	58.60	95.80	28.61	195.70	25.80	301.60
LDH (U/L)	F	<4	204	636.8	216.1	606.5	480.5	751.0	327.1	1172.8	245.0	1536.0
	F	≥4	392	648.4	290.0	600.0	448.5	752.3	278.0	1482.9	220.0	2116.0
	M	<4	207	701.6	311.4	624.0	508.0	793.0	334.0	1485.0	298.0	3115.0
	M	≥4	128	611.1	262.0	568.5	422.0	751.5	235.7	1357.5	204.0	1414.0
TBIL (µmol/L)	F	<4	204	1.38	0.91	1.20	0.70	1.80	0.20	3.69	0.00	6.50
	F	≥4	387	1.39	0.92	1.20	0.70	1.80	0.20	3.80	0.00	6.10
	M	<4	208	1.47	0.76	1.40	0.90	2.00	0.22	3.26	0.10	4.50
	M	≥4	128	1.43	0.76	1.25	0.90	1.80	0.32	3.70	0.20	4.30
Chol (mmol/L)	F	<4	204	2.91	0.90	2.80	2.40	3.24	1.24	5.62	0.99	7.01
	F	≥4	392	2.85	0.90	2.86	2.31	3.28	1.23	4.61	0.66	8.69
	M	<4	207	2.89	0.76	2.74	2.37	3.16	1.81	4.89	1.50	5.67
	M	≥4	128	2.53	0.64	2.42	2.07	2.96	1.43	3.97	1.39	4.72
Cl (mmol/L)	F	<4	203	106.2	2.5	106.5	104.5	107.9	100.9	110.9	97.8	112.3
	F	≥4	383	105.7	3.0	106.0	103.9	107.8	98.1	110.8	90.7	113.5
	M	<4	207	105.8	2.7	105.9	104.2	107.7	100.3	110.5	98.4	114.5
	M	≥4	123	104.6	2.4	104.7	103.2	106.2	99.1	108.9	97.3	109.8
BIC (mmol/L)	F	<4	205	20.68	3.42	21.10	19.00	22.80	12.14	26.74	5.80	27.90
	F	≥4	392	21.88	3.69	22.20	19.90	24.10	13.03	29.02	5.00	32.90
	M	<4	207	21.46	3.44	21.50	19.20	23.80	12.66	28.08	9.70	29.30
	M	≥4	128	23.85	3.80	24.50	21.78	26.60	12.47	30.36	11.50	32.40
Fe (mmol/L)	F	<4	205	19.73	6.32	19.17	15.44	22.37	8.89	35.51	5.98	47.56
	F	≥4	392	19.69	5.90	19.95	16.16	23.19	7.54	30.87	5.79	52.64
	M	<4	207	19.38	5.01	19.60	16.13	22.43	9.94	29.43	4.60	37.63
	M	≥4	128	22.70	5.52	22.97	19.92	25.65	13.29	32.84	1.23	38.48
K (mmol/L)	F	<4	203	3.68	0.35	3.64	3.47	3.86	3.13	4.37	3.06	6.39
	F	≥4	385	3.62	0.37	3.58	3.36	3.87	3.00	4.43	2.68	5.06
	M	<4	208	3.67	0.31	3.63	3.47	3.85	3.19	4.34	2.86	4.99
	M	≥4	123	3.77	0.36	3.76	3.53	4.01	3.11	4.61	3.07	4.71
Na (mmol/L)	F	<4	203	143.8	2.5	143.8	142.2	145.5	138.5	148.6	135.5	150.8
	F	≥4	386	143.1	3.2	143.3	141.2	145.0	137.2	148.2	120.4	151.1
	M	<4	207	144.1	2.3	144.3	142.6	145.7	139.3	148.8	138.6	150.0
	M	≥4	123	144.4	2.4	144.7	143.2	146.0	137.4	147.9	136.8	148.2
Phos (mmol/L)	F	<4	203	1.58	0.39	1.57	1.30	1.85	0.85	2.45	0.77	2.67
	F	≥4	392	1.04	0.35	1.02	0.79	1.23	0.52	1.86	0.37	3.23
	M	<4	207	1.68	0.36	1.68	1.42	1.91	0.99	2.42	0.84	2.77
	M	≥4	128	1.29	0.35	1.28	1.04	1.51	0.58	2.03	0.48	2.76
Ca (mmol/L)	F	<4	204	2.45	0.14	2.45	2.37	2.55	2.15	2.71	1.95	2.77
	F	≥4	392	2.32	0.14	2.31	2.22	2.41	2.01	2.62	1.93	2.74
	M	<4	207	2.44	0.11	2.44	2.35	2.50	2.20	2.68	2.13	2.74
	M	≥4	127	2.37	0.12	2.36	2.29	2.43	2.17	2.68	2.14	2.75
URE (mmol/L)	F	<4	202	7.74	1.54	7.67	6.78	8.75	4.89	10.99	3.13	11.82
	F	≥4	392	6.88	1.75	6.82	5.74	7.91	3.85	11.16	2.65	13.57
	M	<4	207	7.65	1.60	7.58	6.55	8.63	5.00	11.44	3.95	12.52
	M	≥4	128	6.82	1.62	6.75	5.65	7.84	3.97	10.20	3.51	10.63
Cre (mmol/L)	F	<4	203	54.69	10.79	53.70	47.30	60.15	36.47	82.38	31.00	102.40
	F	≥4	391	64.90	13.91	63.50	55.25	73.30	41.12	95.08	23.30	119.60
	M	<4	207	56.70	10.56	55.40	49.70	63.95	36.72	79.60	30.80	92.80
	M	≥4	128	84.67	17.17	81.75	72.18	95.83	55.98	123.13	50.50	129.40

**Table 6 animals-13-00445-t006:** Hematologic reference intervals and percentiles in the cynomolgus macaques.

Labtest (Unit)	Gender	Age	Count	Mean	SD	Median	P25	P75	RefLow	RefUpp	Min	Max
RBC (10^12^/L)	F	<4	209	5.64	0.47	5.65	5.38	5.91	4.59	6.60	3.96	7.08
	F	≥4	393	5.63	0.71	5.56	5.14	6.05	4.37	7.31	3.99	8.32
	M	<4	212	5.64	0.47	5.70	5.34	5.95	4.64	6.51	4.43	7.05
	M	≥4	127	5.61	0.63	5.56	5.19	6.11	4.12	6.82	3.90	6.92
HGB (mmol/L)	F	<4	209	7.79	0.48	7.80	7.50	8.10	6.73	8.78	6.10	8.90
	F	≥4	393	7.44	0.67	7.40	7.00	7.90	5.90	8.70	4.70	9.30
	M	<4	212	7.89	0.45	7.90	7.60	8.20	7.00	8.70	6.90	8.90
	M	≥4	127	7.97	0.46	8.00	7.70	8.30	7.00	8.90	6.80	9.10
HCT (L/L)	F	<4	209	0.38	0.02	0.38	0.36	0.40	0.33	0.42	0.29	0.43
	F	≥4	393	0.38	0.03	0.37	0.36	0.40	0.31	0.44	0.27	0.50
	M	<4	212	0.38	0.02	0.38	0.36	0.40	0.34	0.42	0.33	0.43
	M	≥4	127	0.39	0.02	0.39	0.38	0.41	0.34	0.45	0.33	0.45
MCV (fL)	F	<4	209	67.55	4.30	67.60	64.40	70.20	59.23	76.38	58.00	77.30
	F	≥4	393	67.12	5.87	66.90	63.00	71.20	55.07	78.62	52.40	83.70
	M	<4	212	67.71	4.43	67.85	64.78	70.30	58.90	77.34	56.20	79.30
	M	≥4	127	70.33	6.04	70.60	65.80	74.40	59.40	83.40	58.00	85.60
MCHC (amol)	F	<4	209	1387.9	100.0	1386.0	1329.0	1452.0	1194.3	1591.0	1052.0	1636.0
	F	≥4	392	1333.9	142.2	1326.0	1228.5	1436.3	1069.0	1611.5	1027.0	1723.0
	M	<4	212	1405.1	97.0	1400.0	1354.3	1465.3	1167.0	1605.3	1142.0	1663.0
	M	≥4	127	1435.1	140.8	1437.0	1339.0	1540.5	1172.2	1719.6	1160.0	1769.0
MCHC (mmol/L)	F	<4	209	20.54	0.64	20.60	20.20	21.00	19.23	21.68	17.60	22.20
	F	≥4	392	19.86	0.81	19.90	19.30	20.50	18.28	21.30	16.70	22.10
	M	<4	212	20.76	0.62	20.80	20.40	21.20	19.40	21.87	18.80	22.20
	M	≥4	127	20.40	0.63	20.40	20.00	20.80	19.04	21.60	18.80	21.80
PLT (10^9^/L)	F	<4	209	366.2	94.0	351.0	300.0	426.0	208.3	584.0	130.0	715.0
	F	≥4	392	349.0	90.9	339.0	293.0	402.5	171.1	527.5	58.0	654.0
	M	<4	212	337.7	72.8	332.5	290.0	382.8	181.8	479.0	119.0	537.0
	M	≥4	127	341.7	69.5	343.0	293.0	385.0	222.4	491.4	53.0	533.0
PDW (fL)	F	<4	190	13.67	1.96	13.50	12.10	15.08	10.66	18.02	10.10	19.90
	F	≥4	377	13.24	1.94	13.00	11.80	14.50	10.25	17.66	9.20	21.80
	M	<4	200	13.01	1.89	12.60	11.80	13.93	10.20	17.88	9.70	18.90
	M	≥4	124	12.57	1.38	12.40	11.60	13.13	10.11	15.78	9.80	16.00
MPV (fL)	F	<4	190	11.41	0.94	11.30	10.80	12.10	9.60	13.42	9.40	13.80
	F	≥4	377	11.08	0.96	11.00	10.40	11.70	9.30	13.00	8.80	14.10
	M	<4	200	11.11	0.87	11.00	10.60	11.70	9.50	12.80	9.00	13.30
	M	≥4	124	10.89	0.80	10.90	10.40	11.40	9.30	12.59	9.20	12.70
P-LCR (%)	F	<4	190	36.33	6.99	36.25	31.63	40.98	22.36	50.21	21.10	53.70
	F	≥4	377	34.20	7.44	33.50	29.00	39.20	19.93	48.98	15.60	55.10
	M	<4	200	33.99	6.57	33.10	29.68	38.38	21.04	47.56	18.30	51.50
	M	≥4	124	32.74	6.22	32.65	28.40	36.70	20.21	45.14	19.20	47.20
PCT (%)	F	<4	190	0.423	0.089	0.415	0.360	0.470	0.300	0.629	0.170	0.940
	F	≥4	377	0.388	0.079	0.380	0.340	0.440	0.229	0.546	0.160	0.710
	M	<4	200	0.378	0.070	0.370	0.330	0.420	0.260	0.520	0.160	0.630
	M	≥4	124	0.374	0.061	0.370	0.330	0.413	0.261	0.519	0.260	0.530
WBC (10^9^/L)	F	<4	209	13.62	4.77	12.97	9.99	16.08	6.27	25.48	5.10	33.14
	F	≥4	393	14.05	5.92	12.90	9.82	17.52	5.04	29.02	3.81	39.81
	M	<4	212	12.46	4.62	11.49	9.57	14.02	6.19	26.93	5.31	35.88
	M	≥4	127	11.04	4.19	10.13	8.08	12.68	5.49	21.24	4.93	25.82
Neut (10^9^/L)	F	<4	209	9.19	4.80	8.53	5.78	11.50	2.10	21.01	1.48	30.59
	F	≥4	392	10.85	5.98	9.72	6.24	14.33	2.02	25.65	1.48	36.22
	M	<4	211	8.09	4.81	7.17	5.01	9.79	1.93	22.74	1.28	30.03
	M	≥4	127	7.27	4.30	6.66	3.79	9.43	1.80	17.71	1.37	20.92
Mono (10^9^/L)	F	<4	209	0.611	0.307	0.550	0.400	0.750	0.205	1.478	0.170	2.270
	F	≥4	392	0.707	0.377	0.620	0.468	0.880	0.250	1.522	0.130	3.980
	M	<4	211	0.552	0.234	0.500	0.390	0.665	0.250	1.108	0.160	1.930
	M	≥4	127	0.636	0.289	0.570	0.430	0.785	0.260	1.300	0.110	2.010
Lymph (10^9^/L)	F	<4	209	3.632	1.778	3.330	2.210	4.780	1.213	8.113	0.960	10.270
	F	≥4	392	2.310	1.197	2.000	1.420	2.923	0.803	5.353	0.560	7.740
	M	<4	211	3.642	1.752	3.300	2.450	4.290	1.263	8.616	0.830	12.270
	M	≥4	127	2.922	1.251	2.670	1.980	3.620	1.052	5.636	0.870	8.020
Eo (10^9^/L)	F	<4	209	0.170	0.182	0.120	0.040	0.230	0.000	0.700	0.000	1.070
	F	≥4	392	0.176	0.197	0.110	0.050	0.240	0.010	0.611	0.000	1.640
	M	<4	211	0.140	0.160	0.090	0.040	0.190	0.000	0.689	0.000	1.070
	M	≥4	127	0.201	0.166	0.150	0.080	0.290	0.010	0.644	0.000	0.900
Baso (10^9^/L)	F	<4	208	0.019	0.014	0.020	0.010	0.020	0.010	0.050	0.000	0.140
	F	≥4	389	0.013	0.008	0.010	0.010	0.020	0.000	0.030	0.000	0.060
	M	<4	212	0.016	0.012	0.010	0.010	0.020	0.000	0.047	0.000	0.100
	M	≥4	126	0.010	0.007	0.010	0.010	0.010	0.000	0.030	0.000	0.040

**Table 7 animals-13-00445-t007:** Glucose values in rhesus macaques in mmol/L. In 2018–2019, ketamine was used as a sedation protocol, and in 2020–2021, ketamine combined with medetomidine was used as a sedation protocol.

Year	Gender	Age	Count	Mean	SD	Median	P25	P75	RefLow	RefUpp	Min	Max
2018–2019	F	<4	346	3.80	1.00	3.70	3.23	4.25	2.19	5.85	0.96	10.84
2020–2021	F	<4	222	6.26	2.03	6.09	4.94	7.66	2.80	10.43	1.85	13.04
2018–2019	F	≥4	751	3.59	1.16	3.44	2.98	3.99	1.91	6.07	0.74	17.88
2020–2021	F	≥4	713	6.10	1.85	6.02	4.86	7.21	2.93	9.86	1.14	15.93
2018–2019	M	<4	311	4.22	1.17	4.05	3.47	4.71	2.41	7.29	1.73	9.91
2020–2021	M	<4	185	6.21	1.90	6.13	4.72	7.35	3.07	10.25	2.31	12.26
2018–2019	M	≥4	263	4.19	0.93	4.04	3.60	4.68	2.71	6.56	1.86	8.56
2020–2021	M	≥4	222	6.09	1.67	6.12	4.87	7.17	3.29	9.59	1.62	11.60

**Table 8 animals-13-00445-t008:** Glucose values in cynomolgus macaques in mmol/L. In 2018–2019, ketamine was used as a sedation protocol, and in 2020–2021, ketamine combined with medetomidine was used as a sedation protocol.

Year	Gender	Age	Count	Mean	SD	Median	P25	P75	RefLow	RefUpp	Min	Max
2018–2019	F	<4	83	3.51	1.25	3.29	2.75	3.78	1.51	7.66	1.45	8.76
2020–2021	F	<4	122	6.28	2.07	6.08	4.69	7.57	2.84	11.11	2.65	13.18
2018–2019	F	≥4	192	3.54	2.01	2.99	2.45	3.94	1.53	9.30	1.28	18.84
2020–2021	F	≥4	199	6.01	1.94	5.75	4.68	7.12	2.87	10.33	2.26	13.75
2018–2019	M	<4	104	3.66	0.97	3.58	3.02	4.12	2.05	6.54	1.96	7.01
2020–2021	M	<4	103	6.87	1.87	6.90	5.52	7.98	3.97	11.08	3.52	11.20
2018–2019	M	≥4	81	4.23	1.51	3.91	3.29	4.67	2.37	7.99	1.89	12.82
2020–2021	M	≥4	47	7.04	2.62	6.42	5.51	8.25	3.11	17.65	2.94	19.28

**Table 9 animals-13-00445-t009:** Combined reference intervals for age and gender.

Spec	Lab Tests	Count	Mean	SD	Median	P25	P75	RefLow	RefUpp	Min	Max
rhesus	K (mmol/L)	2976	3.58	0.36	3.56	3.35	3.77	2.95	4.34	1.24	6.38
rhesus	Na (mmol/L)	3016	145.90	2.73	146.10	144.30	147.60	140.30	150.80	127.20	155.00
rhesus	MCHC (amol)	3011	21.22	0.62	21.20	20.90	21.60	20.20	22.20	15.40	33.60
cynomolgus	Eo (10^9^/L)	939	0.170	0.183	0.11	0.05	0.24	0	0.64	0	1.64

**Table 10 animals-13-00445-t010:** Combined reference intervals for age, partitioned per gender.

Spec	Labtest	Gender	Count	Mean	SD	Median	P25	P75	RefLow	RefUpp	Min	Max
rhesus	Cl (mmol/L)	F	2036	107.45	2.80	107.40	105.60	109.30	102.00	112.80	93.20	116.40
rhesus		M	980	106.01	2.73	106.00	104.30	107.80	101.20	111.40	89.50	118.60
rhesus	K (mmol/L)	F	2001	3.55	0.37	3.53	3.33	3.74	2.93	4.32	1.24	6.38
rhesus		M	975	3.62	0.35	3.60	3.39	3.82	3.02	4.38	1.84	5.65
rhesus	PLT (10^9^/L)	F	2036	317.55	74.84	317.00	271.00	364.25	176.00	466.08	55.00	726.00
rhesus		M	971	308.87	73.07	313.00	263.00	357.00	150.30	441.70	69.00	628.00
rhesus	P-LCR (%)	F	1991	34.33	7.22	34.50	29.40	39.50	19.28	47.32	13.50	56.60
rhesus		M	945	32.91	7.27	32.90	28.00	38.30	18.30	46.11	12.50	58.10
rhesus	Neut (10^9^/L)	F	2026	9.30	5.10	8.38	5.22	12.68	2.25	20.61	0.89	32.62
rhesus		M	959	7.06	4.18	5.92	3.94	9.39	1.83	17.55	0.93	25.63
rhesus	Mono (10^9^/L)	F	2037	0.64	0.31	0.58	0.42	0.79	0.23	1.40	0.07	3.01
rhesus		M	973	0.57	0.25	0.53	0.39	0.71	0.21	1.15	0.04	1.68
cynomolgus	TBIL (µmol/L)	F	591	1.38	0.92	1.20	0.70	1.80	0.20	3.72	0.00	6.50
cynomolgus		M	336	1.45	0.76	1.30	0.90	1.90	0.30	3.22	0.10	4.50

**Table 11 animals-13-00445-t011:** Combined reference intervals for gender, partitioned per age.

Species	Brief	Age	Count	Mean	SD	Median	P25	P75	RefLow	RefUpp	Min	Max
rhesus	ALT (U/L)	<4	1069	31.39	12.57	29.50	24.10	35.80	15.10	59.45	6.50	177.90
rhesus		≥4	1953	36.64	28.55	28.80	21.50	41.80	12.09	111.79	4.80	383.80
rhesus	AST (U/L)	<4	1069	41.03	17.36	37.70	31.30	46.70	19.45	81.60	12.40	215.20
rhesus		≥4	1952	31.34	16.13	28.70	23.70	35.20	15.48	61.50	8.60	347.60
rhesus	LDH (U/L)	<4	1067	468.80	191.42	424.00	338.00	554.50	220.70	935.90	151.00	1733.00
rhesus		≥4	1950	428.09	203.53	383.00	295.00	506.75	199.00	859.95	142.00	2807.00
rhesus	Chol (mmol/L)	<4	1066	3.47	0.71	3.37	2.99	3.89	2.39	5.25	0.86	6.35
rhesus		≥4	1950	3.38	0.63	3.36	2.97	3.76	2.15	4.72	0.99	6.20
rhesus	BIC (mmol/L)	<4	1068	22.59	3.71	22.75	20.70	25.00	14.89	29.03	0.80	34.00
rhesus		≥4	1952	23.79	3.82	24.00	21.60	26.10	15.88	30.82	0.10	36.50
rhesus	MCV (fL)	<4	1058	3.51	0.37	3.47	3.30	3.69	2.94	4.35	1.84	6.07
rhesus		≥4	1954	3.61	0.35	3.60	3.39	3.80	2.96	4.34	1.24	6.38
rhesus	MCH (amol)	<4	1057	68.12	2.73	68.20	66.20	69.90	62.75	73.50	56.10	75.50
rhesus		≥4	1954	69.69	2.84	69.60	67.80	71.60	64.29	75.20	58.70	83.30
rhesus	MCHC (mmol/L)	<4	1057	1451.33	61.23	1452.00	1414.00	1492.00	1325.45	1566.55	1166.00	1643.00
rhesus		≥4	1954	1475.19	71.63	1475.00	1434.00	1515.00	1354.00	1592.00	1039.00	2791.00
rhesus	PDW (fL)	<4	1008	21.31	0.56	21.40	21.00	21.70	20.30	22.26	16.30	22.80
rhesus		≥4	1928	21.17	0.65	21.20	20.90	21.50	20.10	22.10	15.40	33.60
rhesus	WBC (10^9^/L)	<4	1058	13.19	2.19	12.80	11.60	14.40	9.82	18.48	9.00	21.80
rhesus		≥4	1954	12.60	1.92	12.30	11.20	13.70	9.50	17.20	8.40	21.00

**Table 12 animals-13-00445-t012:** Combined glucose reference intervals for gender, partitioned per age.

Year	Sex	Count	Mean	SD	Median	P25	P75	RefLow	RefUpp	Min	Max
2018–2019	F	1097	3.65	1.11	3.52	3.06	4.11	1.94	5.98	0.74	17.88
2018–2019	M	574	4.21	1.06	4.05	3.53	4.69	2.50	6.71	1.73	9.91
2020–2021	F	935	6.14	1.89	6.04	4.86	7.30	2.85	9.98	1.14	15.93
2020–2021	M	407	6.15	1.78	6.13	4.82	7.25	3.19	10.02	1.62	12.26

### 3.2. Age versus Gender

Appendix A show the percentage difference (Delta %) in the blood values. Animals < 4 years (young animals) and animals > 4 years (adult animals) were compared in both species along with gender.

In both genders of both species, a significant age effect in the ALP levels was observed. Higher ALP levels were observed in young animals compared to adult animals (27.5% in rhesus females, 15.6% in rhesus males, 28.8% in cynomolgus females, and 25.8% in cynomolgus males (all *p* < 0.000001). Male rhesus adults showed higher ALP levels compared to female rhesus adults (18.9%; *p* < 0.000001). In young male rhesus macaques, the ALP level was higher compared to young female rhesus macaques (5.1%; *p* < 0.000001).

Significantly higher ALAT levels were observed in young cynomolgus males compared with adult males (7.2%; *p* < 0.000001).

AST levels were significantly higher in young rhesus females compared to adult females (6.5%; *p* < 0.000001). When comparing young males and adult males, higher AST levels were noted in both species (6.8% and 8% in rhesus and cynomolgus macaques, respectively; *p* < 0.000001).

GGT levels showed an age and gender effect in both species. In young animals, the levels were significantly higher than in adults (11.7% rhesus females; 6.9% rhesus males; 10.9% cynomolgus females; 7.8% cynomolgus males; *p* < 0.000001), and males had significantly higher levels than females (5.1% young rhesus; 10% adult rhesus; 5.4% young cynomolgus; 8.6% adult cynomolgus macaques; *p* < 0.000001).

The TBIL levels in the rhesus macaques were significantly higher in young females compared to adult females (5%; *p* < 0.000001).

In the rhesus macaques, the Fe levels were higher in adult males compared to young males (6.5%; *p* < 0.000001) and higher in adult males compared to adult females (5.3%; *p* < 0.000001).

The Phos levels of both species were significantly higher in young animals compared to adults in females (11.6% rhesus; 10.7% cynomolgus; *p* < 0.000001) and males (6.2% rhesus; 6.8% cynomolgus; *p* < 0.000001). In adult macaques, the Phos levels were higher in males compared to females (7.6% rhesus; 5.8% cynomolgus; *p* < 0.000001).

The Cre levels were higher in adult rhesus compared to young rhesus macaques (7.4% females; 9.1% males; *p* < 0.000001). In the cynomolgus macaques, we observed higher levels in adult males compared to young males (9.6%; *p* < 0.000001) and also higher levels in adult males compared to adult females (6.3%; *p* < 0.000001).

WBC was significantly higher in young rhesus males compared to adult rhesus males (7.9%; *p* < 0.000001). In both species, a gender effect in adult animals was observed: females showed a higher count than males (7.1% rhesus; 6% cynomolgus; *p* < 0.000001). Neut was higher in females compared to males in both rhesus (11.9%; *p* < 0.000001) and cynomolgus macaques (9.3%; *p* < 0.000001). In rhesus macaques, a higher count in young males compared to adult males (9%; *p* < 0.000001) was observed. Lymphocytes showed a gender effect with a higher count in males compared to females in both species (6.7% rhesus; 7.2% cynomolgus; *p* < 0.000001). In addition, higher levels were observed in young animals compared to adults in both genders (12.1% female rhesus; 7.2% male rhesus, 12.5% female cynomolgus; *p* < 0.000001; 5.3% male cynomolgus (*p* < 0.001). Eo was higher in young female rhesus macaques compared to adult female rhesus macaques (13.6%; *p* < 0.00001). In contrast to that, in cynomolgus macaques, a higher count in adult males compared to young males (12.5%; *p* < 0.00001) was observed. Baso was higher in young female rhesus and young female cynomolgus macaques compared to young males (16.7%; *p* < 0.0001) and adult females (16.7%; *p* < 0.000001).

In 2018–2019 (sedation protocol A), we observed higher glucose levels in adult male rhesus macaques compared to adult female adult rhesus macaques (6.7%; *p* < 0.000001).

### 3.3. Indoor versus Outdoor Housing

Outdoor-housed rhesus macaques had a significantly higher WBC compared to indoor-housed animals (7.6% in females and 5% in males; *p* < 0.000001) due to an increase in neutrophils (12.5% in females and 7.9% in males *p* < 0.000001) (Appendix A). The TBIL levels in outdoor-housed rhesus macaques were higher compared to indoor-housed rhesus macaques (5.6% in females; *p* < 0.0001 and 6.5% in males; *p* < 0.00001).

### 3.4. Weight-for-Height Indices (WHI)

ALP levels of non-overweight adult macaques were significantly higher compared to overweight animals in female (5.3%; *p* < 0.000001) and male (17.1%; *p* < 0.0001) rhesus macaques and also in female cynomolgus macaques (10.7%; *p* < 0.00001) (Appendix A). Higher Chol levels were observed in non-overweight female cynomolgus macaques compared to overweight animals (6.1%; *p* < 0.0001).

### 3.5. Pregnancy

Pregnant animals had lower ALB levels compared to non-pregnant animals in cynomolgus macaques (5.5%; *p* < 0.000001). Moreover, the Chol levels (12.5% in rhesus and 16.4% in cynomolgus macaques; *p* < 0.000001), URE levels (5.6% in cynomolgus macaques; *p* < 0.000001), Lymph levels (6.9% in rhesus macaques; *p* < 0.001), ALP levels (6.2% in rhesus macaques; *p* < 0.0001), and ALT levels (6.9% in rhesus macaques; *p* < 0.00001) were also lower in pregnant animals compared to non-pregnant animals. In 2018–2019 (sedation protocol A), pregnant rhesus macaques had lower glucose levels compared to non-pregnant animals (10.4%; *p* < 0.000001).

The Fe levels (5.2% in rhesus macaques; *p* < 0.00001), WBC (7.7% in cynomolgus macaques; *p* < 0.0001), TBIL levels (9.6% in cynomolgus macaques; *p* < 0.0001), Baso (16.7% in cynomolgus macaques; *p* < 0.001), Neut (11.4% in cynomolgus macaques; *p* < 0.0001), and Mono (10.5% in cynomolgus macaques; *p* < 0.00001) were higher in pregnant animals compared to non-pregnant animals (Appendix A).

## 4. Discussion

Our study included a uniquely high number of animals, and in addition, all data were calculated using observed percentiles without parametric assumptions, as the majority of hematologic and serum biochemical values do not display a true normal (gaussian) distribution [42]. The results of tests for normality presented in our study confirmed this. When data are not normally distributed, the usual summary statistics (means and standard deviations) and “normal ranges” should be used with caution or should not be used at all. Nevertheless, overall, the obtained reference intervals were comparable with those found in earlier studies [27,28,29,32,33,34,35] and were within the physiological range of our previously accumulated BPRC hematologic and serum biochemical reference intervals.

Nowadays, captive macaques in breeding colonies are maintained socially in outdoor enclosures [39,43,44,45,46,47]. In the United States of America, already, seventy-five percent of rhesus macaques at national primate research centers are housed in outdoor enclosures [47]. However, this housing type is not always possible for reasons such as climate or research programs (e.g., Animal Biosafety Levels 3 and 4). Animals are then housed indoors, in small groups or pairs. The major benefit of outdoor enclosures is exposure to seasonal fluctuations in light and climate and increased sensory stimulation, which provide great opportunities for exploration and manipulation that all contribute positively to the animals’ welfare. However, outdoor housing results in varying background incidences of primary and opportunistic bacterial, viral, parasitic, and fungal pathogens. The soil in the outdoor enclosures can form a host of microbes and parasites. The animals also might have direct or indirect contact with wildlife such as rodents and birds, which can result in exposure to various microorganisms. In addition, indoor facilities are cleaned once per week, but outdoor enclosures are ‘cleaned’ less frequently. Furthermore, cleaning and disinfection in indoor facilities are more efficient than in outdoor enclosures. Therefore, our finding that animals with outdoor access have higher WBC counts than indoor-housed animals is explainable. Eosinophilia is a central feature of the host response to parasitic infection, but it is generally not observed in protozoal infections. Therefore, it is a more specific sign of helminth infection [48,49]. In helminth infections, the eosinophilia is usually most pronounced early in infection, coinciding with the larval migration through tissues, which then slowly decreases over time. The observed eosinophil levels in our study did not show a clinically relevant difference between indoor and outdoor housing.

Several blood parameters are known to demonstrate significant changes with age in primates [16,50,51]. ALP, as a byproduct of osteoblast activity, is generally higher in children due to rapid bone metabolism and growth. In our study, ALP demonstrated the greatest increase with age, which is in line with the literature, in which it is described that bone growth in young primates produces elevated ALP levels [15,17,18,20,50,51,52,53].

In non-overweight adult female and male rhesus macaques and female cynomolgus macaques, the ALP showed clinically relevant higher levels compared to overweight animals. Our finding of lower Chol levels in overweight female cynomolgus macaques is similar to Yue et al. (2016) who described lower total Chol levels in captive cynomolgus macaques with overweight compared to lean controls [54]. In addition, in a study with a smaller sample size the Chol levels were described as not being associated with WHI in adult female cynomolgus macaques [55]. In humans, being overweight tends to increase Chol levels in the blood, which is in contrast to our findings in macaques [56].

Physiological changes in pregnancy and puerperium are principally influenced by hormonal changes. Many hematological shifts that occur during these periods are physiological and are of inconsequential concern [57,58]. Our finding that WBC is increased during pregnancy is similar to human data [59]. However, it was observed that a significant increase in the numbers of neutrophils and a decrease in lymphocytes occurs on day 20 of the pregnancy compared to the non-pregnant state in rhesus macaques [57]. Due to the limited number of pregnancies, we did not divide into trimesters.

In macaques, chemical restraint is required to perform reliable physical examinations, diagnostic exercises, and the collection of tissue or body fluid samples (e.g., cerebrospinal fluid). Sedation not only reduces stress on the animal but promotes safety and improves the quality of data collection. Ketamine hydrochloride is the most used drug for the chemical restraint of macaques. However, ketamine sedation is known to cause significant alterations to serum biochemical and hematological variables [14,23,24,25,30,31,42]. Increased AST, LDH, and CK levels are indicative of local myotoxicity of the injected ketamine, especially when sedations are performed on sequential days. Moreover, a study demonstrated that ketamine sedation reduces leukocyte counts in cynomolgus macaques [31]. Further, reduced leukocyte counts have been observed in rhesus macaques with ketamine sedation alone [23,24]. Ketamine-medetomidine anesthesia is a preferred anesthetic regimen for animals because of its wide safety range and favorable effects on hemodynamics. However, a negative aspect of this regimen is the development of hyperglycemia. α-2 agonists, such as medetomidine, not only impair insulin release from the pancreatic β cells but also increase glucagon release from the α cells. The decrease in the insulin/glucagon ratio results in decreased glucose uptake and increased glucose production by the liver [60]. This is reflected in our results, where we have shown a marked increase (~10%) in glucose levels after medetomidine was introduced to the sedation protocol. This iatrogenic hyperglycemia can hinder the correct interpretation of glucose levels in these animals.

No clinically relevant gender-related differences have been observed for RBC, HGB, and HCT. This is similar to what is described in other reports [3,9,54]. However, others have reported that male macaques show significantly higher RBC, Hb, and HCT values compared to females [11,12,28,32,36,42], similar to observations in humans. Although the general belief is that these results are related to menstrual blood loss in females, scientific justification was not presented.

Climate could be an influential factor in our reference intervals. Wild living primates can face (relative) scarcity and seasonal fluctuations of food resources in their habitat. Macaques adopt active foraging strategies, relying on a variety of food species and adjusting flexibly their food choices based on food availability [61,62]. However, captive macaques are provided a daily balanced diet, resulting in minimal to no variation of intake in nutritional value during the year. Parasite burdens are known to vary seasonally in wildlife, and rainfall is one key aspect of seasonality that has been linked to parasitism in a range of systems. Rainfall can have immediate effects on parasitism rates by affecting parasite survival and movement in the environment, or it can have delayed effects by affecting host susceptibility to parasites through changes in the host body condition or immune function [63,64,65,66,67]. However, the Netherlands has a maritime climate, with no temperature extremes in summer and winter and the absence of a marked wet and dry season (www.weatherbase.com, accessed on 18 January 2023), concluding that it was unlikely that climate was an influential variable for our study.

In the Appendix A, multiple outliers can be observed, the impact of these outliers is minimized because we employed non-parametric statistical methods to establish reference intervals. Hypothetically, this could be subclinical diseased animals as during the annual health evaluation, the veterinarian noted no clinical abnormalities. The animals with clinical abnormalities were excluded from our analysis. No specific follow-up was indicated for those outliers as all animals were observed twice daily by caretakers.

Our updated reference intervals in macaques demonstrate the importance of using appropriate statistical procedures, homogenous animal populations, and establishing uniform and validated animal housing and husbandry conditions to improve the reproducibility of experiments involving blood sampling. It is imperative to have a reliable and comprehensive set of hematologic and serum biochemical reference intervals based on gender, age, weight-for-height indices, pregnancy, sedation protocol, and housing condition to determine the overall health status of an individual macaque or a colony used as a breeding population.

## 5. Conclusions

Knowing the effect of age, gender, weight-for-height indices, pregnancy, housing conditions, and sedation protocol on hematologic and serum biochemical values in macaques will help us to distinguish the boundaries between physiological and pathological changes. In addition, this knowledge will advance future research to identify other variables that may affect those parameters.

## Figures and Tables

**Table 1 animals-13-00445-t001:** Included number of animals by gender and species.

Gender	Rhesus Macaque	Cynomolgus Macaque
Female	712	193
Male	422	148
Total	1134	341

**Table 2 animals-13-00445-t002:** Number of repeats by gender and species.

Repeats	Female Rhesus Macaques	Male Rhesus Macaques	Female Cynomolgus Macaques	Male Cynomolgus Macaques
1	120	132	17	25
2	100	91	32	57
3	229	122	49	49
4	263	77	95	17

## Data Availability

Data are available on reasonable request.

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
