# Peer review of "Reference Intervals and Percentiles for Hematologic and Serum Biochemical Values in Captive Bred Rhesus (*Macaca mulatta*) and Cynomolgus Macaques (*Macaca fascicularis*)"

_animals, 2023, doi:10.3390/ani13030445_

Round 1

Reviewer 1 Report

Comments on the manuscript “Reference intervals and percentiles for hematologic and serum biochemical values in captive bred rhesus (Macaca mulatta) and cynomolgus macaques (Macaca fascicularis)” submitted to the Animals

General comments

I appreciate the opportunity to review this interesting manuscript. Congratulations to the authors for the laborious study. The authors describe the hematological and biochemical profiles of two captive primate species, the cynomolgus, and the rhesus macaques. The independent variables were age, gender, weight‐for‐height indices, two different sedation protocols, and indoor or outdoor housing conditions. I think that another categorical variable that is forgotten in the text emerged, that is, obese and non-obese individuals, which I will comment on later.

The manuscript is a relevant contribution to primatology and other correlated scientific fields. Despite there being many articles on hematology and biochemical parameters of cynomolgus and the rhesus macaques, the article is original due to the sample size, comparison levels, and modern analytical methods.

In order to better understand the research protocol, further clarification is needed in some parts of the manuscript. For example, there may be fluctuations in blood parameters due to seasonality (Marchal et al., 2012; Setchell et al., 2006), particularly in outdoor animals. Were all phlebotomies performed at the same time of year (summer, winter, etc)? Were samples collected over contiguous day periods?

There is no discussion of blood values considering weight-for-height indices. This discussion must be in the manuscript.

In Figure S2, many outliers can be noticed, with very different values. How did the authors treat the outlier data?

The authors do not discuss the significant differences between obese and non-obese Rhesus females in terms of ALP, AST, GGT, Cl, Phos, URE, RBC, HGB, HCT, MCV, PDW, MPV, P-LCR, WBC, and Neut scores. Also not discussed are the ALP and MCV values for obese and non-obese Rhesus males. Thus, a discussion about the differences found in Cynomolgus monkeys between obese and non-obese females is also expected; and between obese and non-obese males

Other suggestions are below:

Line 64: What is the “AAALAC International”?

Line 180: The concentration (mg/Kg) of drugs for the sedation protocol is determined a priori. However, the animals' weight is determined after sedation. How was the weight of the animals previously estimated to inject the correct volume of sedative?

Lines 347-354: “To provide optimal…potentially confounding data.” This sentence is a relevant justification, but it does not seem to be connected with this part of the discussion. This sentence justifies the work presented, so I suggest that it be included in the introduction.

Line 377: The claim that leukocytosis is not an indicator of infection in monkeys seems unreasonable to me. There can be a lot of variation in white blood cell concentrations, caused by stress, nutritional conditions, and infections. Leukocytosis (or leukopenia) are a very important clinical indicator in primates. For example, in Sudan Virus Infection in Rhesus (Zumbrun et al., 2012); infection by Burkholderia pseudomallei (Yeager et al., 2012); Ebola virus infection (Jaax et al., 1996), etc. Furthermore, the claim that concentrations of up to 30 X 109/L can be detected is based on a book from 52 years ago [49]. At that time, there was a refinement of hematological analyzes and many articles do not corroborate that old finding.

Line 412: The authors do not discuss the lack of gender difference found in this study compared to other studies. It is necessary to discuss this result.

REFERENCES

Jaax, N. K., Davis, K. J., Geisbert, T. J., Vogel, P., Jaax, G. P., Topper, M., & Jahrling, P. B. (1996). Lethal experimental infection of rhesus monkeys with Ebola-Zaire (Mayinga) virus by the oral and conjunctival route of exposure. Archives of Pathology and Laboratory Medicine, 120(2), 140-155.

archal, J., Dorieux, O., Haro, L., Aujard, F., & Perret, M. (2012). Characterization of blood biochemical markers during aging in the Grey Mouse Lemur (Microcebus murinus): impact of gender and season. BMC veterinary research, 8(1), 1-10.

Setchell, J. M., Tshipamba, P., Bourry, O., Rouquet, P., Wickings, E. J., & Knapp, L. A. (2006). Hematology of a semi-free-ranging colony of mandrills (Mandrillus sphinx). International Journal of Primatology, 27(6), 1709-1729.

Yeager, J. J., Facemire, P., Dabisch, P. A., Robinson, C. G., Nyakiti, D., Beck, K., ... & Pitt, M. L. M. (2012). Natural history of inhalation melioidosis in rhesus macaques (Macaca mulatta) and African green monkeys (Chlorocebus aethiops). Infection and immunity, 80(9), 3332-3340.

Zumbrun, E. E., Bloomfield, H. A., Dye, J. M., Hunter, T. C., Dabisch, P. A., Garza, N. L., ... & Nalca, A. (2012). A characterization of aerosolized Sudan virus infection in African green monkeys, cynomolgus macaques, and rhesus macaques. Viruses, 4(10), 2115-2136.

Reviewer 2 Report

 The mansucript analysis the data of hematologic and serum biochemical values in captive bred rhesus (Macaca mulatta) and cynomolgus macaques (Macaca fascicularis) between 2018 and 2021. The results are helpful to veterinarians and researchers. However, the overall innovation of this article needs to be improved, and the analysis of each parameter is relatively rough and simplistic.  for example, the table 3A-3B is descriptive and must to be in-depth statistics and analysis according to the animals, gender and ages. In particular, individual parameters need to be listed according to the functional and physiological significance of the reflection.  In adition, the titles of all tables are lengthy and not refined and need to be modified.

Round 2

Reviewer 2 Report

The author has revised manuscripts.